# Type I Interferon Receptor Subunit 1 Deletion Attenuates Experimental Abdominal Aortic Aneurysm Formation

**DOI:** 10.3390/biom12101541

**Published:** 2022-10-21

**Authors:** Takahiro Shoji, Jia Guo, Yingbin Ge, Yankui Li, Gang Li, Toru Ikezoe, Wei Wang, Xiaoya Zheng, Sihai Zhao, Naoki Fujimura, Jianhua Huang, Baohui Xu, Ronald L. Dalman

**Affiliations:** 1Department of Surgery, Stanford University School of Medicine, Stanford, CA 94305, USA; 2Department of Emergency Medicine, Saiseikai Central Hospital, Minatoku, Tokyo 108-0073, Japan; 3Center for Hypertension Care, Shanxi Medical University First Hospital, Taiyuan 030012, China; 4Department of Physiology, Nanjing Medical University, Nanjing 210093, China; 5Department of Vascular Surgery, Central South University Xiangya Hospital, Changsha 410008, China

**Keywords:** abdominal aortic aneurysm, type I interferon receptor, leukocytes, angiogenesis

## Abstract

Objective: Type I interferon receptor signaling contributes to several autoimmune and vascular diseases such as lupus, atherosclerosis and stroke. The purpose of this study was to assess the influence of type I interferon receptor deficiency on the formation and progression of experimental abdominal aortic aneurysms (AAAs). Methods: AAAs were induced in type I interferon receptor subunit 1 (IFNAR1)-deficient and wild type control male mice via intra-infrarenal aortic infusion of porcine pancreatic elastase. Immunostaining for IFNAR1 was evaluated in experimental and clinical aneurysmal abdominal aortae. The initiation and progression of experimental AAAs were assessed via ultrasound imaging prior to (day 0) and days 3, 7 and 14 following elastase infusion. Aneurysmal histopathology was analyzed at sacrifice. Results: Increased aortic medial and adventitial IFNAR1 expression was present in both clinical AAAs harvested at surgery and experimental AAAs. Following AAA induction, wild type mice experienced progressive, time-dependent infrarenal aortic enlargement. This progression was substantially attenuated in IFNAR1-deficient mice. On histological analyses, medial elastin degradation, smooth muscle cell depletion, leukocyte accumulation and neoangiogenesis were markedly diminished in IFNAR1-deficient mice in comparison to wild type mice. Conclusion: IFNAR1 deficiency limited experimental AAA progression in response to intra-aortic elastase infusion. Combined with clinical observations, these results suggest an important role for IFNAR1 activity in AAA pathogenesis.

## 1. Introduction

Type I interferons (IFNs) are cytokines produced by plasmacytoid dendritic cells and other immune cells following exposure to antigenic stimuli including bacteria, viruses, autoantigens and tumors [1,2]. For example, Type 1 IFN expression is increased in patients with the SARS-CoV-2 virus (COVID-19) [3,4,5,6,7]. Most type I IFNs signal through the type I IFN heterodimer receptor subunit (IFNAR) 1/IFNAR2 to initiate intracellular signaling cascades and type 1 IFN-regulated gene expression [1]. While critical for host defense, dysregulated type I IFN/IFNAR signaling is also associated with autoimmune diseases such as lupus and type 1 diabetes [8,9,10].

In vascular diseases, type I IFNs promote atherosclerosis by altering the functional phenotypes of immune cells, vascular endothelial and smooth muscle cells (SMC) [11]. Genetic deficiency or antibody inhibition of IFNAR1 reduced cerebral infraction volume in a mouse ischemic stroke model [12]. IFNAR1 deficiency also protected against hypoxia-induced pulmonary arterial hypertension [13].

Abdominal aortic aneurysm (AAA) is a chronic life-threatening inflammatory disease characterized by progressive enlargement of the infrarenal aorta leading to rupture and potentially sudden death. Both innate and adaptive immune responses potentially contribute to AAA pathogenesis. Pharmacological depletion or genetic deficiency of macrophages, mast cells, neutrophils and B cells suppressed experimental AAAs [14,15,16,17]. Genetic deficiency or antibody neutralization of IFN-γ, interleukin (IL)-17 or tumor necrosis factor-α attenuated experimental AAAs [18,19,20], as did treatment with transforming growth factor-β, IL-10 and IL-19 [14,21,22,23]. However, the role of IFNAR1 activity in AAA pathogenesis has not been previously investigated.

In the present study, both human AAA surgical specimens and mice deficient for IFNAR1 were employed to examine the role of type I IFN activity in aortic aneurysm pathogenesis.

## 2. Materials and Methods

### 2.1. Experimental AAA Modeling

IFNAR1-deficient (IFNAR1^−/−^, B6.129S2-*Ifnar1*^tm1Agt^/Mmjax, MMRC strain #032045-JAX) and wild type (WT, strain #000664) mice on a C57BL/6J background were obtained from the Jackson Laboratory, Bar Harbor, Maine, USA and housed at the Stanford University Research Animal Facility (Stanford, CA, USA). IFNAR1^−/−^ mice display no remarkable phenotypic anomalies by the age of 6 months [24]. AAAs were induced in male mice at 10–12 weeks of age by transient intra-infrarenal aortic infusion of porcine pancreatic elastase (PPE) as previously described [25,26]. Briefly, following median laparotomy, isolation and control of the infrarenal aorta, an aortotomy was created via a 30-G needle. Thirty microliters of PPE in phosphate-buffered saline (Type 1, 1.5 units/mL, catalog #E-1250-100MG; Sigma-Aldrich, St. Louis, MO, USA) was infused via a syringe pump (Model 100, KD Scientific, Holliston, MA, USA). Following PPE infusion, the aortotomy and laparotomy were closed with 10–0 nylon (Microsurgery Instruments Inc, Bellaire, TX, USA). and 6–0 silk (Ad Surgical, Sunnyvale, CA, USA) sutures, respectively. Mice were recovered and housed in separate cages with free access to water and food. The mouse PPE infusion model produces a fusiform, focal AAA, independent of serum lipid levels or systemic blood pressure, that recapitulates both clinical and pathological features of human AAA disease except for the absence of intraluminal thrombus and progression to rupture [27,28,29,30]. All procedures were performed under sterile conditions in compliance with Stanford Laboratory Animal Care Guidelines and approved by the Stanford University Administrative Panel on Laboratory Animal Care (protocol #11131).

### 2.2. Immunohistochemistry for IFNAR1

Non-aneurysmal human abdominal aortae were obtained from two organ donors. Infrarenal human AAA specimens were obtained from patients undergoing open surgical repair (n = 6). Collection and use of these specimens were approved by the Human Subject Research Review Board at Xiangya Hospital, Central South University School of Medicine, Changsha, Hunan, China. Non-aneurysmal and aneurysmal mouse aortae were prepared via intra-infrarenal aortic infusion of phosphate-buffered saline (non-aneurysmal) and PPE (aneurysmal), respectively (n = 3 mice/group). All human and mouse aortae were fixed with 4% paraformaldehyde, embedded in paraffin and sectioned (4 μm). Sections were stained with a rabbit anti-IFNAR1 polyclonal antibody (ab244357, Abcam, Waltham, MA, USA) or purified normal rabbit IgG (AB-105-C, R&D Systems, Minneapolis, MN, USA), and detected with mouse- and rabbit-specific HRP (house radish peroxidase)/DAB (3,3′-diaminobenzidine) detection kits (ab64264, Abcam, Waltham, MA, USA) as described previously [22,31]. IFNAR1 expression was quantified as the percentage of positive staining area with the total aortic cross-sectional (ACS) area using Image J Fuji software (Ver 2.0.0-rc-43/1.53m).

### 2.3. In Vivo Assessment of AAA Formation and Progression

Serial transabdominal ultrasonography was employed to monitor AAA formation and progression. Although controversy exists regarding the optimal measurement criteria for clinical AAA diameter (e.g., external to external vs. anterior to anterior [32,33]), resolution limitations of mouse transabdominal ultrasonography at 40 MHz make definition of aortic mural boundaries and wall thickness impractical. Thus, for the present study, AAAs were defined and monitored as a function of luminal aortic diameters. As noted previously, this model does not accumulate aortic mural thrombus. Transverse maximal infrarenal aortic diameters were measured prior to (baseline or day 0) and days 3, 7 and 14 following PPE infusion using the Vevo 2100 ultrasound system (Visualsonics, Toronto, ON, Canada) as previously described [30,34]. An AAA was defined as a ≥50% increase in aortic diameter over the baseline.

### 2.4. Histological Analyses

Acetone-fixed frozen aortic sections were prepared from mice 14 days after PPE infusion and used for Verhoeff’s Van Gieson (EVG) stain and immunohistochemistry as described previously [29]. Monoclonal antibodies (mAb) for immunohistochemistry were biotinylated anti-SMC α actin (Clone 1A4 from Thermo Fisher Scientific Inc, Waltham, MA, USA as well as CD68 (Clone FA-11 for macrophages), CD4 (Clone GK 1.5), CD8 (Clone 53-5.8), B220 (Clone RA3-6B2 for B cells) and CD31 (Clone 390 for blood vessels) from Biolegend Inc, San Diego, CA, USA. Biotinylated anti-rat IgG (Catalog number 112-065-006) and streptavidin-peroxidase conjugate (Catalog number 016-030-084) were obtained from Jackson ImmunoResearch Inc., West Grove, PA, USA. Peroxidase substrate AEC (3-amino-9-ethylcarbazole) kits were purchased from Vector Laboratories, Burlingame, CA, USA. Aortic macrophage accumulation, medial elastin degradation and SMC loss were scored as grade I (mild) to IV (severe) using previously reported histological grading criteria [29]. Other subsets of leukocytes and neovessels were quantitated as the number of leukocyte subset-antibody positive cells and CD31-positive vessels per ACS, respectively [29]. All histological assessments were analyzed by a single experienced experimental pathologist who was blinded to experimental group assignment.

### 2.5. Data Analysis

All continuous variables were tested for normal distribution using the Shapiro–Wilk test. When normally distributed, data were reported as mean and standard deviation (SD), with the Student’s t-test or repeated measures two-way analysis of variance followed by Newman–Keuls post-test used to test statistical significance between groups. If not normally distributed, data were reported as median and 25% and 75% interquartile intervals, with the nonparametric Mann–Whitney test used to test statistical difference between groups. The difference in experimental AAA incidence was tested by the Log-Rank test. All analyses were performed using Prism (Ver 9.0; GraphPad Software LLC, San Diego, CA, USA). Significance was determined at the *p* < 0.05 level.

## 3. Results

### 3.1. IFNAR1 Expression in Experimental and Clinical AAAs

To assess whether IFNAR1 expression is altered in aneurysmal aortae, non-aneurysmal and aneurysmal aortic sections obtained from mice and patients were immuno-stained using a rabbit anti-IFNAR1 polyclonal antibody. Although some staining was present in the media of non-aneurysmal mouse aortae (Figure 1A, PBS infusion), transmural IFNAR1 stain was dramatically increased in experimental AAAs (Figure 1B, PPE infusion), with a significantly larger IFNAR1 stain-positive area in aneurysmal (0.8 ± 0.2: positive staining/total ACS area) vs. non-aneurysmal (0.3 ± 0.1) aortae (*p* < 0.05, 3 mice/group). Normal rabbit IgG as the negative control antibody revealed no staining over the background in either aneurysmal or non-aneurysmal aortae (Figure 1C,D).

In human organ donor specimens, no or rare IFNAR1 stain was detected in non-aneurysmal aortae (Figure 2A,B). In sections obtained from aneurysmal aortae in AAA patients, positive transmural IFNAR1 staining was readily apparent (Figure 2C,H). Positive staining was most pronounced in the adventitia, particularly within leukocyte aggregates (mean and SD: 2.9 ± 2.2) vs. the media (0.2 ± 0.1) (*p* < 0.05, 6 patients/group). No staining was seen on non-aneurysmal organ donor (not shown) or aneurysmal (Figure 2I) aortae using control rabbit IgG. Thus, IFNAR1 expression appears increased in both experimental and clinical AAAs.

### 3.2. Attenuated AAA Formation and Progression in IFNAR1^−/−^ Mice

To test the hypothesis that INFAR1 activity contributes to experimental AAA pathogenesis, aneurysms were induced in male IFNAR1^−/−^ and age-matched WT mice. Baseline aortic diameter was similar between IFNAR1^−/−^ and WT mice (Figure 3A,B). Following PPE infusion, a time-dependent, progressive aortic enlargement was observed in both groups (Figure 3A,B). However, reduced enlargement was noted in IFNAR1^−/−^ mice at all time points following AAA induction, with the final aortic diameter of 1.04 ± 0.08 and 1.24 ± 0.09 mm in IFNAR1^−/−^ and WT mice, respectively (Day 14, Figure 3A,B). AAAs, defined by a ≥50% increase in aortic diameter over the baseline level, formed in only 7 of 12 IFNAR1^−/−^ (58.3%) vs. all WT mice following PPE infusion (Figure 3C). Thus, genetic IFNAR1 deficiency was associated with reduced formation and progression of experimental AAAs.

### 3.3. Attenuated Medial Elastin Degradation and Smooth Muscle Cell Depletion in IFNAR1^−/−^ Mice

Medial elastolysis and SMC depletion are characteristic features of AAA pathogenesis. To assess the influence of IFNAR1 deficiency on medial integrity, we performed aortic EVG and SMC-α actin staining following PPE infusion (Figure 4). In WT mice, both medial elastin and SMCs were significantly degraded and depleted, respectively, by 14 days following PPE infusion. However, genetic deficiency of IFNAR1 was associated with significant medial elastin and SMC retention. Thus, IFNAR1 deficiency enhanced medial integrity in the PPE-induced experimental AAA model.

### 3.4. Attenuated Mural Leukocyte Accumulation in IFNAR1^−/−^ Mice

To evaluate the influence of IFNAR1 activity on mural leukocyte cellularity in experimental AAAs, we stained aortic sections with leukocyte subset-specific mAbs at sacrifice (Figure 5). In WT mice, macrophages, identified as CD68^+^ cells, accounted for most leukocytes, followed by CD4^+^, CD8^+^ and B220^+^ cells, respectively. In contrast, AAAs created in IFNAR1-deficient mice demonstrated reduced accumulation of all leukocyte subsets, reflecting a more than 90% reduction compared to WT mice. These results confirm that genetic IFNAR1 deficiency was associated with attenuated leukocyte accumulation in experimental AAAs.

### 3.5. Attenuated Mural Angiogenesis in IFNAR1^−/−^ Mice

Mural neoangiogenesis is an additional characteristic morphologic feature of clinical and experimental AAAs that may promote mural instability, diameter enlargement and rupture. As seen in Figure 6, dense mural neovessels (39.3 ± 10.6 vessels/ACS), as identified by CD31 antibody immunostaining, were present in WT AAAs. However, in AAAs created in IFNAR1^−/−^ mice, neovessel density was 7.1 ± 2.6 vessels/ACS, a >80% reduction relative to WT mice (*p* < 0.01). These results indicate that IFNAR1 deficiency was associated with attenuated aneurysmal angiogenesis.

## 4. Discussion

IFNAR1 expression was increased in experimental and clinical AAAs. IFNAR1 deficiency was associated with reduced incidence and progression of experimental AAAs following aneurysm induction. In histological analyses, IFNAR1 deficiency was associated with relative preservation of medial elastin and SMC density, reduced mural macrophage, T and B cell infiltration and neoangiogenesis. These findings suggest a causal or enhancing role for IFNAR1 activation in AAA pathogenesis.

Plasmacytoid dendritic cells were previously noted to be present in experimental and clinical AAA, expressing IFN-α, an avid ligand for IFNAR1 [35]. Plasmacytoid dendritic cells were mobilized rapidly from bone marrow into peripheral blood during experimental AAA formation and progression [36]. Pharmacological interventions such as antibody-mediated plasmacytoid dendritic cell depletion and type 1 IFN signaling inhibition were effective in suppressing experimental AAAs in conjunction with reduced aortic T cell accumulation [35,36]. The current findings in IFNAR1-deficient mice complement the previous observations and reinforce the significance of IFNAR1 activity in AAA pathogenesis.

Reduced aortic mural leukocyte accumulation and neoangiogenesis noted in IFNAR1-deficient mice are consistent with the hypothesized role of leukocytes and angiogenesis in AAA pathogenesis. Pharmacological depletion or genetic deficiency of macrophages, B cells, CD4^+^ or CD8^+^ T cells suppressed experimental AAAs [14,17,18,37,38]. Anti-angiogenesis strategies including VEGF-A sequestration/neutralization, VEGF receptor 2 inhibition and alternative angiogenic inhibitors suppress, whereas proangiogenic interventions such as VEGF-A supplementation augmented, experimental AAAs [25,31,39,40,41]. Conversely, pharmacological stabilization of the proangiogenic transcription factor hypoxia inducible factor-1α reversed diabetes-related AAA suppression in experimental AAAs [42,43].

Type I IFNs and IFNAR1-mediated signaling induce immune responses to bacterial and viral infection, implying a potential relationship between AAA disease and pathogen-driven immune responses. Respiratory and genitourinary tract, skin, abdominal cavity and even bloodstream bacterial infections are all associated with aortic aneurysms, with an odds ratio of 1.2–3.1 after adjusting for major co-morbid risk factors and medication regimen [44]. HIV patients are also at higher risk, with an adjusted odds ratio of 4.5 [45]. Cytomegalovirus gene UL75 is also more frequently detected in aortae of AAA patients compared to healthy controls [46]. However, antimicrobial therapeutic trials to suppress early AAA disease (e.g., roxithromycin [47,48] and azithromycin [49]) have proven inconclusive or ineffective, suggesting that infection alters disease progression through immune responses (including type I IFN signaling) rather than the microbiological properties of the inciting pathogen itself.

Perhaps most importantly in this regard, the COVID-19 pandemic, caused by the SARS-CoV-2 pathogen, has affected more than 96 million Americans as of 19 October 2022 (https://covid.cdc.gov/covid-data-tracker). In a national commercial laboratory COVID-19 seroprevalence study, anti-SARS-CoV-2 nucleocapsid antibody, an index for SARS-COV-2 infection, was detected in nearly 50% and 33% of people aged 50–64 and >65 years, respectively—the population susceptible to AAA disease [50]. Long-term COVID-19 syndrome affects multiple organ systems, including the cardiovascular system, even in fully vaccinated individuals [51,52,53]. In COVID-19 patients, the type 1 IFN response is augmented in asymptomatic and mildly infected patients as well as the early phases of severe disease as compared to late phase or critically ill patients [3,4,5,6,54].

COVID-19 may increase clinical AAA risk [55,56]. COVID-19 patients under surveillance for smaller AAAs experienced unexpectedly rapid aneurysm enlargement regardless of sex, baseline diameter or other traditional risk factors [57,58,59,60,61,62]. Conversely, vaccination with inactivated SARS-CoV-2 vaccine attenuates type 1 IFN response [3]. Thus, understanding the consequences of vaccination and post-acute SARS-CoV-2 infection on aneurysm enlargement, rupture risk and/or need for emergent surgical repair in convalescent COVID-19 patients will improve overall care and, potentially, outcomes for patients with AAA disease [55]. Additionally, a potential critical question is whether vaccination with the United States Food and Drug Administration (US FDA)-approved mRNA and adenovirus vaccines will limit further progression of existing AAAs in convalescent COVID-19 patients.

Type 1 IFNs are closely linked to autoimmune diseases [9]. AAA risk was increased in patients with autoimmune diseases such as lupus and psoriasis [63,64,65,66,67], and anti-rheumatic therapies were associated with reduced rates of AAA enlargement [68]. While autoantigens and other autoimmunity components may contribute to AAA disease [69,70], the current findings linking IFNAR1 to experimental AAA pathogenesis provide further insights into the positive association between autoimmune diseases and aortic aneurysms.

IFNAR1/IFNAR2 heterodimer receptor activation promotes sequential phosphorylation of tyrosine kinase/STAT2 and Janus-activated kinase/STAT1 to trigger type 1 IFN-mediated inflammatory responses [38]. Small molecule inhibitors to tyrosine kinase or Janus-activated kinase, including baricitinib, tofacitinib, ruxolitinib, upadcitinib, fedratinib and oclacitinib, are indicated for treatment of certain autoimmune diseases and malignant tumors [71]. The humanized anti-IFNAR1 antagonist mAb, anifrolumab, was recently approved by the US FDA for treatment of systemic lupus erythematosus [72]. These results add to a growing body of literature suggesting that agents targeting type I IFN-mediated inflammatory responses, depending on their therapeutic index and potential efficacy in this application, may be of value in clinical AAA disease management.

In conclusion, IFNAR1 appears to play a role in experimental aneurysm formation and progression. Pharmacological strategies targeting IFNAR1 or the type 1 IFN-mediated proinflammatory pathway may enhance medical regimens for AAA disease management.

## Figures and Tables

**Figure 1 biomolecules-12-01541-f001:**
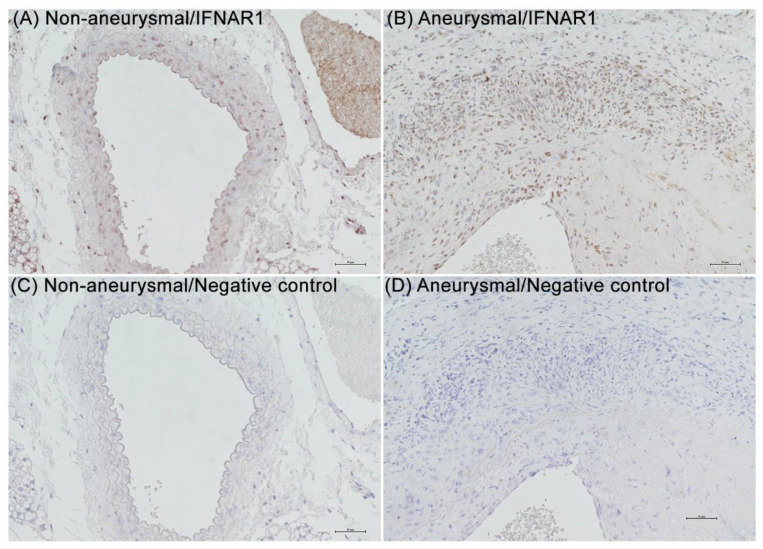
**Immunostaining of type I IFN receptor subunit 1 in non-aneurysmal and aneurysmal mouse aortae.** Fourteen days following intra-infrarenal aortic elastase (aneurysm group) or phosphate-buffered saline infusion (non-aneurysm group), the involved aorta was harvested, fixed with 10% formalin, embedded in paraffin and sectioned (4 μm). Sections were stained with a rabbit anti-human type I interferon receptor subunit 1 (IFNAR1) polyclonal antibody (cross-reacts with mouse IFNAR1) or negative control antibody (normal rabbit IgG) via the immunoperoxidase procedure and visualized with peroxidase substrate 3,3′-diaminobenzidine. (**A**,**B**) Representative IFNAR1 staining from non-aneurysmal (**A**) and aneurysmal (**B**) aortae. (**C**,**D**) No staining with normal rabbit IgG. These results were reproduced in 3 mice in each group. Scale bar: 50 μm.

**Figure 2 biomolecules-12-01541-f002:**
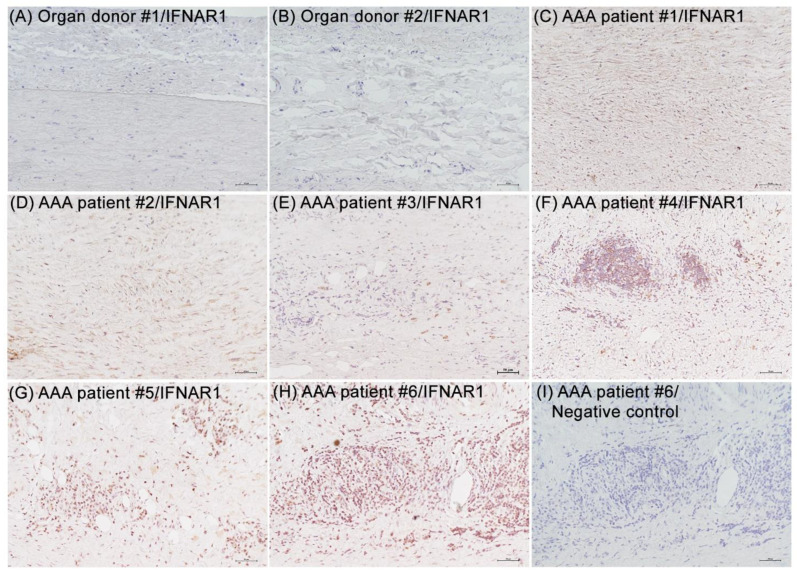
**Immunostaining of type I IFN receptor subunit 1 in non-aneurysmal and aneurysmal human aortae.** Formalin-fixed paraffin sections were prepared from aortic specimens harvested at organ donation (n = 2) or AAA repair (n = 6), stained with a rabbit anti-human type I interferon receptor subunit 1 (IFNAR1) polyclonal antibody or negative control antibody (normal rabbit IgG), and visualized with peroxidase substrate 3,3′-diaminobenzidine. (**A**,**B**) Rare or no IFNAR1 staining detected in organ donor aortae. (**C**–**H**) IFNAR1 staining was localized in the media and adventitia (within and outside of lymphocyte aggregates) of AAA specimens. (**I**) No staining noted in aneurysmal (**I**) and non-aneurysmal organ donor aortae (not shown) with negative control antibody. Scale bar: 50 μm.

**Figure 3 biomolecules-12-01541-f003:**
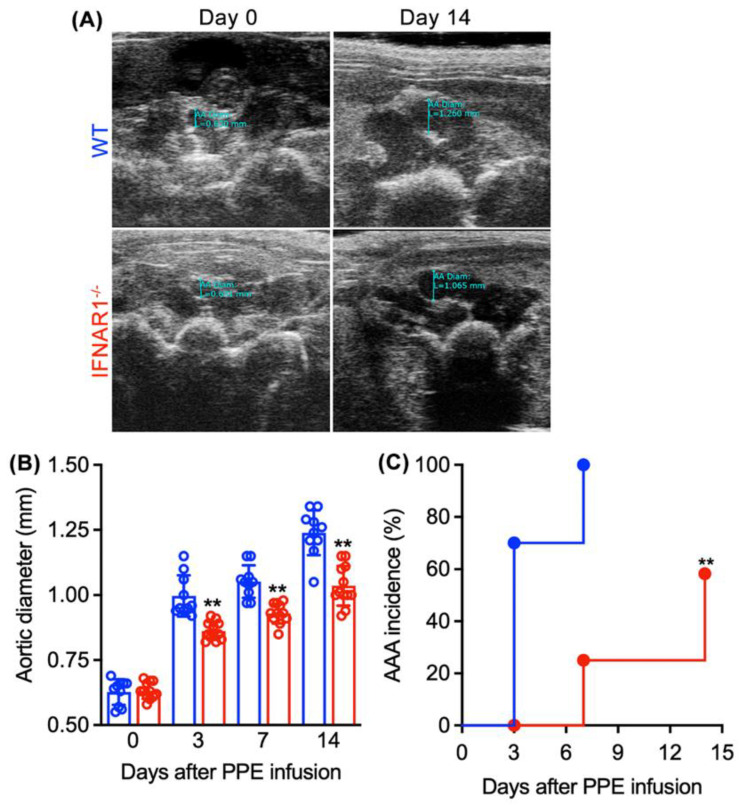
**Type I IFN receptor subunit 1 deficiency attenuates the formation and progression of experimental AAAs.** Male wild type (WT, n = 10) and type I interferon receptor subunit 1-deficient (IFNAR1^−/−^, n = 12) mice on C57BL/6J genetic background received intra-aortic porcine pancreatic elastase (PPE) infusion for AAA induction. Maximal transverse aortic diameters were measured at baseline and indicated days following PPE infusion via transabdominal ultrasonography. (**A**) Representative images from IFNAR1^−/−^ and WT mice at baseline and 14 days following PPE infusion. (**B**) Mean and standard deviation (SD) of aortic diameters at baseline and days 3, 7 and 14 after PPE infusion. Two-way ANOVA followed by two group comparison, ** *p* < 0.01 compared to WT mice at the same time point. (**C**) AAA incidence. An AAA was defined as a more than 50% increase in the aortic diameter over the baseline. ** *p* < 0.01 compared to WT mice by log-rank test.

**Figure 4 biomolecules-12-01541-f004:**
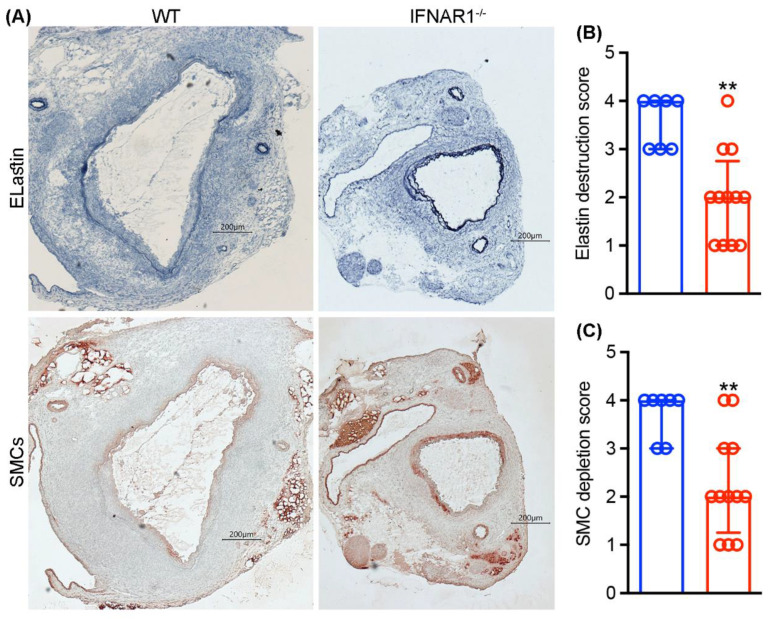
**Type 1 IFN receptor subunit 1 deficiency reduces medial elastin degradation and smooth muscle cell depletion in experimental AAAs.** Acetone-fixed aortic frozen sections were prepared from wild type (WT, n = 7) and type 1 IFN receptor subunit 1-deficient (IFNAR1^−/−^, n = 12) mice 2 weeks after elastase infusion and stained using the Verhoeff’s Van Gieson (EVG) procedure and immunostaining for medial elastin and smooth muscle cells (SMCs), respectively. Elastin degradation and SMC depletion were graded as I (mild) to IV (severe). (**A**) Representative histologic images demonstrate relative medial elastin and SMC preservation in IFNAR1^−/−^ as compared to WT mice. (**B**,**C**) Quantification (median and interquartile) of medial elastin and SMC destruction scores between groups. Nonparametric Mann–Whitney test, ** *p* < 0.01 compared to WT mice. Scale bar: 200 μm.

**Figure 5 biomolecules-12-01541-f005:**
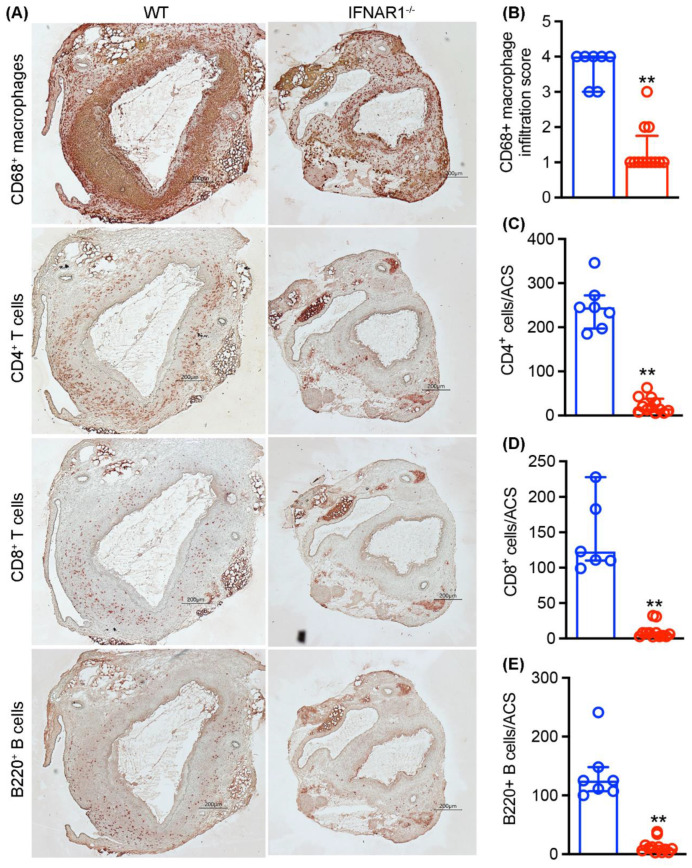
**Type I IFN receptor subunit 1 deficiency reduces aortic leukocyte accumulation in experimental AAAs.** Aortic frozen sections from elastase-infused wild type (WT, n = 7) and type 1 IFN receptor subunit 1-deficient (IFNAR1^−/−^, n = 12) mice were stained with monoclonal antibodies against CD68 for macrophages, CD4 for CD4^+^ T cells, CD8 for CD8^+^ T cells and B220 for B cells. Macrophage accumulation was scored from I (mild) to IV (severe). Other leukocytes were enumerated as positively stained cells/aortic cross-section (ACS). (**A**) Representative images for macrophages, CD4^+^ and CD8^+^ T cells and B cells in elastase-infused WT and IFNAR1^−/−^mice. (**B**–**E**) Quantification (median and interquartile) of mural macrophages, CD4^+^ T cells, CD8^+^ T cells and B cells between groups. Nonparametric Mann–Whitney test, ** *p* < 0.01 compared to WT mice. Scale bar: 200 μm.

**Figure 6 biomolecules-12-01541-f006:**
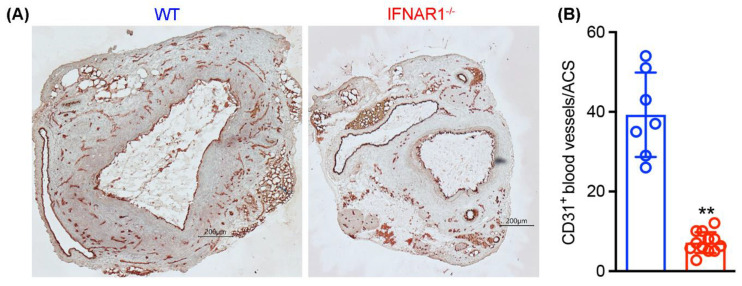
**Type I IFN receptor subunit 1 deficiency is protective against mural angiogenesis in experimental AAAs.** Aortic frozen sections from elastase-infused wild type (WT, n = 7) and type I IFN receptor subunit 1-deficient (IFNAR1^−/−^, n = 12) mice were stained with an anti-mouse CD31 monoclonal antibody to identify neovessels. (**A**) Representative CD31 immunohistochemical images for neoangiogenesis in AAAs from WT and IFNAR1^−/−^ mice. (**B**) Mean and standard deviation of mural neoangiogenesis quantified as the number of CD31-positive vessels per aortic cross-section (ACS). Student t’s test, ** *p* < 0.01 compared to WT mice. Scale bar: 200 μm.

## Data Availability

All data were included within the article.

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
