# Peer review of "Type I Interferon Receptor Subunit 1 Deletion Attenuates Experimental Abdominal Aortic Aneurysm Formation"

_biomolecules, 2022, doi:10.3390/biom12101541_

Round 1
Reviewer 1 Report
The authors investigated the role of type I interferon receptor subunit 1 (IFNAR1) in abdominal aortic aneurysm formation and progression. By using a mouse model of AAA, authors demonstrated that IFNAR1 deletion in mice attenuated aneurysm initiation and progression, with reduced infiltrating inflammatory cells, and preserved medial elastin and smooth muscle cell cellularity. The manuscript is well written.
Minor points:
1) Each title for the Results should be consistent.
2) Fig.1B, what was the unit for stain positive area? Similarly, Fig. 2C-2H, what was the unit for positive staining of leukocyte aggregates?
3) Fig. 2B label (Organ donor #1/IFNR1) should be (Organ donor #2/IFNR1).
Reviewer 2 Report
In this study, T. Shoji et al. investigated role of Type I interferon receptor (IFNAR) in abdominal aortic aneurysm (AAAs) formation and progression using IFNAR1 deficiency mouse model. Authors observed that progression of AAAs was substantially attenuated, along with diminished medial elastin degradation, smooth muscle cell depletion, leukocyte accumulation and neo-angiogenesis, in IFNAR1 deficient mice. They concluded that IFNAR1 deficiency limited experimental AAA progression in response to intra-aortic elastase infusion. This study suggests a regulatory role for IFNAR1 activity in AAA pathogenesis. Overall, this is a well-designed study using a unique mouse model with extensive histo-pathology analysis. Data is convincing and conclusion is solid. In addition, this study is timely in terms of its potential linkage to COVID-19, as COVID-19 may increase clinical AAA risk because the type 1 IFN response is augmented in asymptomatic and mildly COVID-19 infected patients. My major comments are:
(1) Authors should discuss the role of IFNAR signaling in inflammatory aneurysm versus atherosclerotic aneurysm. Likely, IFNAR signaling is primarily associated with inflammatory aneurysm.
(2) Authors are suggested to add some background information about the type 1 IFN response in COVID-19, so the discussion can well echo the introduction.
Reviewer 3 Report
This is a manuscript addressing “Type I interferon receptor subunit 1 deletion attenuates experimental abdominal aortic aneurysm formation”. This is well-written and the conclusions are based on the results. The reviewer has some minor concerns need to be addressed
1) The phenotype of IFNAR1 deficient (IFNAR1-/-) mice would affect the pathology of AAA. Some information about the general appearance, blood pressure, and other factors which might have influence on the aorta would be informative.
2) The PPE infusion model might induce very focal AAA which is different from clinical feature of AAA. More information about the diffuse pathology of this model would be necessary.
3) The methods for immunohistochemistry are separated in two paragraphs: “Immunohistochemistry for IFNAR1” and “Histological analyses”. These may be combined together.
4) Aortic cross-section (ACS) should be explained when it appears for the first time.
5) The inner aortic diameter was measured in this study; do the authors have the data of outer diameter because it would be usual methods in clinical settings. Also, the outer diameter might reflect the inflammation in the aortic wall better inner one.
6) Figure 2. Immunostaining of type I IFN receptor subunit 1 in human AAAs: the title should be revise because it includes the normal aorta from donors.
7) Figure 6: the indication with some arrows for angiogenesis would be informative.
